# Treatments Targeting the Androgen Receptor and Its Splice Variants in Breast Cancer

**DOI:** 10.3390/ijms25031817

**Published:** 2024-02-02

**Authors:** Amy H. Tien, Marianne D. Sadar

**Affiliations:** 1Canada’s Michael Smith Genome Sciences Centre, BC Cancer, Vancouver, BC V5Z 1L3, Canada; 2Department of Pathology and Laboratory Medicine, University of British Columbia, Vancouver, BC V6T 1Z7, Canada

**Keywords:** androgen receptor, breast cancer, combination therapy, AR-V7 splice variant, targeted therapy

## Abstract

Breast cancer is a major cause of death worldwide. The complexity of endocrine regulation in breast cancer may allow the cancer cells to escape from a particular treatment and result in resistant and aggressive disease. These breast cancers usually have fewer treatment options. Targeted therapies for cancer patients may offer fewer adverse side effects because of specificity compared to conventional chemotherapy. Signaling pathways of nuclear receptors, such as the estrogen receptor (ER), have been intensively studied and used as therapeutic targets. Recently, the role of the androgen receptor (AR) in breast cancer is gaining greater attention as a therapeutic target and as a prognostic biomarker. The expression of constitutively active truncated AR splice variants in breast cancer is a possible mechanism contributing to treatment resistance. Therefore, targeting both the full-length AR and AR variants, either through the activation or suppression of AR function, depending on the status of the ER, progesterone receptor, or human epidermal growth factor receptor 2, may provide additional treatment options. Studies targeting AR in combination with other treatment strategies are ongoing in clinical trials. The determination of the status of nuclear receptors to classify and identify patient subgroups will facilitate optimized and targeted combination therapies.

## 1. Introduction

The first case of breast cancer was recorded approximately 3500 years ago in 1600 BC, when the ancient Egyptians noted lumps spreading across the breast [1]. Today, female breast cancer is the most commonly diagnosed cancer worldwide, with approximately 2.26 million cases [2]. Fortunately, the survival rate for breast cancer has improved primarily due to early detection and more effective treatment options. The 5-year relative survival rate for female breast cancer was approximately 89% in the United States from 2012 to 2018. However, the relative survival rate drops to only 28% if the disease disseminates to form distant metastases [3]. This poor survival rate emphasizes the need for better treatment options for metastatic disease.

Current treatments for breast cancer include surgery, chemotherapy, radiation, and hormone and targeted therapies. The efficacies of each treatment or combination treatment depend on the stage and type of breast cancer. Most breast cancer is driven by the ligand-activated transcription factor, estrogen receptor-alpha (ERα), which mediates the effects of the sex hormone, estrogen. The current targeted hormone therapies for breast cancers that express ERα involve the inhibition or modulation of ERα transcriptional activity by selective ERα modulators (SERMs such as tamoxifen and raloxifen), aromatase inhibition (e.g., anastrozole, exemestane, and letrozole) and ERα degradation (e.g., fulvestrant). The acquired resistance to these therapies may develop after prolonged treatments due to changes that may include alterations in the tumor microenvironment and the reduced reliance of the cancer on the expression of ERα [4].

In the search for the development of alternative targeted therapies for breast cancer, it was discovered that the closely related androgen receptor (AR) can be detected in the majority of all types of breast cancers [5]. The AR mediates the biological effects of androgens, which include testosterone and dihydrotestosterone (DHT). Elevated levels of tissue concentrations of DHT were measured in breast carcinoma and ductal carcinoma in situ (DCIS) and were in the range of 110–698 pg/g tissue and 140–1593 pg/g, respectively [6]. These concentrations of DHT are comparable to those measured in recurrent prostate cancer and are at levels that can transactivate the AR [7]. The AR has been a therapeutic target in prostate cancer for approximately six decades. There are a vast number of clinically approved therapeutic options that target AR, thereby facilitating a rapid translation of any of these treatments for clinical testing in breast cancer patients. However, to date, there is no approved AR targeting therapy for breast cancer, which is largely considered to be due to inadequate patient selection criteria for clinical trials. In addition, currently, the clinical significance of AR expression and its biological functions, as well as the reliance on cross-talk between the AR, ERα, and other molecules, remains controversial [8,9].

## 2. Breast Cancer Types

The complex connections among different steroid hormone receptor signaling pathways contribute to the high heterogeneity in breast cancer cells. ERα and the progesterone receptor (PR) are involved in the growth and development of breast cancer. Based on the molecular expression of hormone receptors and human epidermal growth factor receptor 2 (HER2), breast cancer can be separated into four major subgroups: (1) luminal A that is positive for ERα and PR and without HER2 amplification; (2) luminal B that is positive for both ERα, PR, and amplified HER2; (3) HER2-amplified cancer that is negative for ERα; and (4) basal-like cancer in which the majority of the cases, approximately 75–80%, are triple-negative for ERα, PR and HER2 and referred to as triple-negative breast cancer (TNBC). The remaining cases of basal-like cancer are positive for both ERα and/or HER2.

Male breast cancer (MBC) is rare and accounts for less than 1% of all breast cancer patients [10,11]. MBC has a greater frequency in black African American and Israeli males [12] and transwomen [13]. MBC is predominately positive for the expression of ERα, PR, and AR and negative for HER2 [14,15,16]. Here, we focus on female breast cancer.

## 3. AR Structure and Splice Variants

### 3.1. Steroid Hormone Receptor Family

Targeting steroid hormone receptor signaling has been a therapeutic strategy in many endocrine-related cancers and other diseases. This family of receptors includes the AR (NR3C4), ER (ESR1; NR3A1 and ESR2; NR3A2), PR (NR3C3), glucocorticoid receptor (GR; NR3C1), and mineralocorticoid receptor (MR; NR3C2). This family of receptors has similarities in their structures, with each member having three functional domains and a flexible hinge region. Since these receptors are ligand-activated transcription factors, they have a DNA-binding domain (DBD), which is the region of the protein that interacts with DNA. Their ligand-binding domain (LBD) is located at the C-terminus of the protein and contains activation function-2 (AF-2). Upon ligand binding, these receptors are transactivated and translocated to the nucleus, and then they bind to hormone response elements on the DNA to regulate transcriptional activity. The hinge region connects the DBD and LBD and includes the nuclear translocation signal. The N-terminal domain (NTD) is highly variable in length and amino acid sequence between steroid hormone receptors, but all NTDs across this family are predominantly intrinsically disordered. The NTD contains the AF-1 region. AF regions play important roles in recruiting multiprotein coregulatory complexes, which regulate transcriptional activity.

### 3.2. AR Structure

Human AR has approximately 920 amino acids, but this may vary depending on polymorphisms and the lengths of repetitive amino acid tracts within its NTD. The *AR* gene includes eight canonical exons on the X chromosome (locus: Xq11–Xq12). The full-length AR is decoded from exons 1–8. There are also at least seven cryptic exons that contribute to various splice variants of AR (AR-Vs). Some AR-Vs that lack an LBD are androgen-independent because they are constitutively active in the absence of androgen [17,18,19,20].

#### 3.2.1. AR-NTD

The AR-NTD comprises the majority of the protein by being approximately 557 amino acids long and is decoded from exon 1. The AR-NTD is predominantly intrinsically disordered, with approximately 13% helical secondary structure, and this helical content may increase upon binding to interacting proteins [21,22]. The flexibility in conformation that intrinsically disordered regions impart allows for high specificity and low-affinity interactions with coregulatory proteins and the transcriptional machinery [21]. The majority of AR transcriptional activity is mediated by its NTD through its AF-1 region [23]. Within this region, there are two transactivation units (tau): Tau-1 (amino acid residue 101–370) and Tau-5 (amino acid residues 360–485) [23]. These two transactivation units interact with the transcriptional machinery to regulate AR transcriptional activity.

#### 3.2.2. AR-LBD

The AF-2 region within the AR-LBD is unique from other steroid hormone receptors in that it does not contribute to much, if any, transcriptional activity for the AR. Crystal structure analyses have revealed that the folded AR-LBD consists of 11 alpha-helices and 2 antiparallel beta-sheets, to create a ligand-binding pocket [24]. Androgen binding alters its conformation to reposition helix 12 over its ligand-binding pocket. This change in conformation allows the AF-2 region to have N/C interactions (interaction between AR-NTD and AR-LBD) [25]. N/C interactions (1) provide a binding interface on AF-2 for coregulators to contribute to the regulation of transcriptional activity; (2) slow the off-rate of the ligand from the ligand-binding pocket; (3) lengthen the half-life of the AR protein; and (4) stabilize the AR on DNA [26,27]. In the absence of a ligand, the LBD negatively regulates the transcriptional activity of the NTD. This has been demonstrated by the deletion of AR-LBD, which results in the constitutive activation of the AR [23], which is analogous to the clinically relevant constitutively active truncated AR-V7 splice variant (described below).

#### 3.2.3. AR-DBD

The AR-DBD has a helical structure. Its three helices consist of two zinc fingers that contain the P-box and D-box, which are essential for binding to DNA and AR dimerization, respectively [28,29]. Mutations within this domain are almost always deleterious to the AR’s transcriptional activity. The AR-DBD is highly similar in sequence identity to the DBDs of other members in this family, thereby making this domain a challenging therapeutic target to achieve specificity to AR. For example, AR-DBD has 80% sequence homology to PR-DBD and 77% with GR and MR.

### 3.3. AR Splice Variants (AR-Vs)

Truncated AR proteins were reported in different diseases such as breast cancer, androgen insensitivity syndrome, and complete testicular feminization in the 1990s [30,31,32,33,34]. The first naturally occurring AR-V was reported in 2005, when it was detected in human placental RNA [35]. This AR-V was named AR45 because the molecular weight of the expressed protein was about 45 kDa. AR45 lacks an NTD, and its mRNA is detected in different types of human tissues, including breast and prostate tissues. AR45 inhibits the transcriptional activity of the full-length AR in the presence or absence of ligands as measured using an AR-driven luciferase reporter assay. Interestingly, AR45 was able to interact with AR-NTD in a mammalian two-hybrid assay, and in the presence of cofactors, it can stimulate transcriptional activity [35]. In subsequent years, many additional AR-Vs were discovered and reported in clinical prostate cancer samples or prostate cancer cell lines. About 20 AR-Vs have been identified. Almost all of the subsequently discovered AR-Vs have an NTD and DBD but lack an LBD in full or in part [8,36]. The lack of LBDs in constitutively active AR-Vs is one of the resistant mechanisms underlying the current therapies that target AR signaling pathways [37]. Among all the AR-Vs, AR-V7 is the most clinically relevant variant. AR-V7 contains only an NTD, a DBD, and a unique sequence of 16 amino acids decoded from the cryptic exon 3b at its C-terminus [19,38]. The expression of AR-V7 has been detected in breast cancer. AR-V7 in breast cancer may be associated with immune function and cell mobility [39].

## 4. AR in Breast Cancer

### 4.1. Expression of AR

AR expression has been reported in the range of 53% to 99% of all breast cancer and 20–40% of TNBC, depending on the study cohort and sensitivity of detection methods (reviews in [5,40,41,42]). In breast cancer cells, AR may have roles in either cell proliferation (stimulatory effect) or antiproliferation (inhibitory effect), depending on the level of ERα expression and disease stages. In recent years, targeting AR in patients with TNBC has been of increasing interest in translational research and clinical trials. This interest is based on a poorer clinical outcome for TNBCs that express AR [43] and that targeting AR reduces the growth of some subtypes of TNBC [42,44,45]. AR antagonists, such as bicalutamide and enzalutamide, or the ablation of androgen production have been investigated in clinical trials for breast cancer [42,46].

AR-V7 mRNA was detected in about 50% of primary breast cancer samples, with the highest expression in the HER2-amplified subgroup [39]. In addition to AR-V7, the transcripts of other variants such as AR-V3, AR-V9, AR-V13, AR-V15, and AR-V18 have been detected in primary breast cancer. Another study measured AR-V7 in about 10% of clinical samples that included primary, recurrent, and metastatic cancers [47]. Approximately 15% of the HER2-amplified subgroup expressed AR-V7, whereas about 18% of the TNBC subgroup expressed AR-V7. Notably, about 40% of the AR-V7-positive breast cancer samples showed an apocrine morphology.

AR variants can be detected in circulating tumor cells (CTCs). CTCs that tested positive for AR-V7 were detected in 41% of patients (9 out of 22) whose breast cancers were positive for ERα [48]. Another study investigating CTCs in TNBC revealed that 27% of patients tested positive for full-length AR, with AR-V7 coexpressed in 73% of these patients [49]. Interestingly, patients with CTCs that expressed AR-V7 and prostate-specific membrane antigen may have worse outcomes after chemotherapy.

### 4.2. AR Role in ERα-Positive Breast Cancer

The analysis of 19 published studies revealed AR expression in about 75% of ERα-positive breast cancer and approximately 32% of ERα-negative breast cancer [50]. In ERα-positive breast cancer, higher levels of nuclear AR protein are usually correlated with improved outcomes and better survival regardless of treatments. The prognostic value of AR in ERα-positive breast cancer has been demonstrated in many studies.

In primary breast cancer tissue, the expression of AR was detected in about 73% of the samples from 413 cases analyzed and was associated with expression of ERα, lower histologic grade, and smaller tumor size, thereby suggesting that the AR is a positive prognostic marker in this group of breast cancer [51]. Meta-analyses of published data have been employed to assess the prognostic significance of the expression of AR in breast cancer. In ERα-positive breast cancer, AR positivity (expression at both mRNA and protein levels) was significantly correlated with improved disease-free survival (DFS) and overall survival (OS), and tumors with an AR:ERα positivity ratio of >0.87 had the best outcome. However, this correlation between AR positivity and DFS or OS was not found in ERα-negative breast cancer [5,52].

One limitation of these meta-analyses that has been discussed is the cut-offs used to determine AR positivity between different studies. Some studies used 1% as the cut-off point, while others used 10%, or even up to 75%. The study by Ricciardelli and colleagues suggested that a 78% cut-off for AR positivity is required to predict the survival outcome for ERα-positive patients [5]. A study by Tagliaferri et al. supports the claim that a high expression of AR is a favorable prognostic indicator of clinical outcome for early-stage ERα-positive/PR-negative/HER2-negative breast cancer [53]. Their study found that ERα-positive breast cancer patients with less than 80% AR expression had a higher risk of relapse than patients with more than 80% AR expression, and these patients with higher AR expression had lower nuclear grade and lower proliferative properties (e.g., lower Ki-67) measured in their tumors. Another study revealed the same correlation/association between AR expression and clinical outcomes in ERα-positive breast cancer. However, improved survival was only observed in patients in the first 5–10 years after diagnosis. Unfortunately, the survival rate became worse beyond 10 years [54].

Another limitation of studies that examine the expression of AR using IHC can be drawn from the types or sources of AR antibody (Ab). Most of the IHC studies that have evaluated the expression of AR protein have used the AR441 antibody (DAKO), which binds to the AR-NTD and cannot distinguish between the full-length AR and AR-Vs. For prostate cancer patients, the expression of AR-Vs alters the treatments that are recommended. Clinical studies have determined that taxanes improve OS compared to treatment with antiandrogens and inhibitors of the AR signaling pathway for prostate cancer patients who test positive for AR-V7 [55,56,57]. In Table 1, the antibodies against AR and AR-Vs are listed.

### 4.3. Elevated Levels of Androgen in Breast Cancer

Androgens, testosterone, and DHT are physiological ligands for the AR. Testosterone is reduced to DHT by 5α-reductase enzymes. In breast tissue, it is the 5α-reductase type 1 isoform that is primarily responsible for the conversion of testosterone to DHT [61]. DHT has a much stronger affinity for AR compared to testosterone. DHT concentrations are elevated (three-fold higher) in tissues of DCIS and breast carcinoma compared to non-neoplastic tissues [62]. The expression levels of 5α-reductase are elevated, and its activity is 4–8 times higher in breast carcinoma tissues than in non-neoplastic tissues [63,64]. The expression levels of specific isoforms of 5α-reductase have been correlated to lymph node metastases and shorter OS of breast cancer patients, with levels of 5α-reductase 1 negatively correlated to histological grade and tumor size [63,65].

The levels of DHT or testosterone are also elevated (2.3-fold higher) in breast cancer tissues and breast cancer models in response to aromatase inhibitors [66,67]. Aromatase is the enzyme that converts testosterone to estrogens. The gene expression analyses of breast cancer tissue from patients neoadjuvant treated with the aromatase inhibitor exemestane revealed that approximately one-half of the 610 androgen-induced genes examined were increased in response to blocking aromatase activity [67]. Consistent with these data, the expression levels of the androgen-induced gene, *KLK3*, or prostate-specific antigen were increased in aromatase-resistant breast cancer tissues [68]. AR expression also increases with neoadjuvant treatment with aromatase inhibitors [69]. Together, these studies point to an altered hormone milieu in response to treatments and emphasize the need to examine the levels of the enzymes involved in the androgen pathway in addition to the levels of AR proteins.

### 4.4. Conflicting Consequences of Cross-Talk between ERα and AR

There is evidence of cross-talk between the AR and ERα to facilitate the growth of some breast cancers [70]. In ERα-positive breast cancer cell lines, estrogen stimulation leads to the nuclear translocation of AR and its binding to unique DNA sites that are enriched in estrogen response elements and that overlap with ER-binding sites. The nuclear translocation of AR was observed in ERα-positive breast cancer cells but not in ERα-negative breast cancer cells. The ERα-driven proliferation in response to estrogen was reduced by the inhibition of AR nuclear translocation with the AR antagonist enzalutamide or by a reduction in the expression of AR with shRNA. Enzalutamide blocked the growth of cell lines and patient-derived xenografts that expressed both AR and ERα. In vivo studies also revealed that enzalutamide had efficacy against tamoxifen-resistant xenografts and reduced the metastatic burden. This study suggests that the inhibition of AR might be an effective treatment for some patients with ERα-positive/AR-positive breast cancer.

On the other hand, a study by Hickey and colleagues demonstrated that AR behaves as a tumor suppressor in ERα-positive breast cancer [71]. They showed that AR transactivation with androgen inhibited ER-driven cell proliferation in an ex vivo patient-derived explant model. Using breast cancer cell lines, they also showed that AR was detected at 42% of estrogen-stimulated ER-binding sites on chromatin when both AR and ER were activated. This suggests that AR could directly affect ER transcriptional activity by redistributing ER. Interestingly, the binding or recruitment of coactivators p300 and SRC-3, which are required for ER signaling, were both reduced and replaced by AR upon AR activation. This led to the repression of ER-regulated cell cycle genes and the inhibition of tumor cell proliferation. Thus, AR activation suppressed ER signaling in ERα-positive breast cancer cells. In this case, the activation of AR, rather than its inhibition, would be the more suitable treatment option for patients with ERα-positive breast cancer when AR behaves as a tumor suppressor. These data support the application of androgenic compounds such as the nonsteroidal selective AR modulator (SARM) enobosarm, which decreases the growth of some ERα-positive breast cancers [72], and they form the rationale for the clinical testing of SARMs for the treatment of some breast cancers.

### 4.5. AR Roles in ER-Negative Breast Cancer

AR-positive/ER-negative breast cancers are classified as molecular apocrine subtypes and include the HER2-amplified and TNBC subgroups. In the HER2-amplified subgroup, AR and HER2 are able to cross-talk. AR activation enhances HER2 expression and increases AR binding to target genes such as *FOXA1*, *XBP1*, *TFF3*, and *KLK3* [73,74]. Cross-talk between AR and HER2 enables them to coregulate their own gene expression. AR activity impacts HER3, which forms a heterodimer with HER2 to activate PI3K (phosphoinositide 3-kinase)/AKT signaling, which leads to cell proliferation [75]. Although AR expression was not significantly associated with DFS (HR 1.20; 95% CI, 0.86–1.69, *p* = 0.28), OS (HR 1.50; 95% CI, 1.01–2.22 *p* = 0.04) was worse in the HER2-positive subgroup according to the analyses performed by Bozovic-Spasojevic et al. [52]. Therefore, AR expression may be useful for the prognosis of OS for patients with ER-negative and HER2-positive breast cancer [76]. Moreover, the inhibition of AR by antiandrogens such as enzalutamide or by shRNA reduced the cell growth of HER2-positive breast cancer cell lines [77]. Together, these data suggest an oncogenic role of AR in the ER-negative and HER2-amplified subgroups.

Patients with TNBC usually have larger and more aggressive tumors, leading to poor clinical outcomes [42,44]. Although TNBC patients respond to chemotherapy, they commonly develop distant recurrence and metastases [78]. TNBC tumors account for 10–20% of all breast cancers, and they are classified into four subgroups based on their tumor-specific gene expression profile, namely basal-like 1, basal-like 2, mesenchymal, and luminal AR (LAR). These four subtypes respond differently to similar neoadjuvant chemotherapy [78,79]. The lack of molecular targets for therapeutic purposes is a challenge for the clinical management of TNBC. The potential of targeting AR may be a feasible option for some TNBC patients.

The LAR subgroup represents a range of 11–22% of TNBC depending on the population studied and analysis methods and is classified based on the luminal gene expression pattern [78,80]. The LAR subgroup is particularly sensitive to antiandrogens. In this subgroup, AR behaves as an oncogenic driver for tumor cell proliferation, and tumors have a high expression of *FOXA1*, *XBP1*, and *KRT18* [78].

FoxA1 is a pioneer factor that opens the chromatin to allow nuclear receptors such as AR to bind to DNA-binding sites [81]. A study by Robinson et al. showed that about 98% of AR binding to DNA events overlapped with the FoxA1-binding region in the molecular apocrine MDA-MB-453 breast cancer cell line, leading to the suggestion that all AR transcriptional activity in molecular apocrine breast cancer cells might be mediated by FoxA1 [82]. In clinical samples from AR-positive nonmetastatic TNBC patients, 42% of patients have tumors coexpressing AR and FoxA1, while only 16% of patients have tumors expressing AR but not FoxA1 [83]. OS was significantly worse for those patients with AR-positive/FoxA1-positive tumors (5-year OS rate was 76.6%) compared to patients with AR-negative tumors (5-year OS rate was 84.8%).

LAR TNBC also has a higher frequency of activating *PIK3CA* mutation than AR-negative TNBC [84]. The inhibition of both AR and PI3K in LAR xenografts decreases tumor growth. Interestingly, the inhibition of PI3K or mTOR resulted in reduced tumor growth in patient-derived xenograft models of LAR that were resistant to AR antiandrogens [85]. These findings suggest that the association between AR and PI3K/Akt/mTOR signaling pathways may contribute to the proliferation of LAR cells. Clinical trials in TNBC have tested, or are currently testing, combinations of AR antagonists with PI3K inhibitors (NCT02457910 [86] and NCT03207529).

## 5. AR as a Target in Monotherapy

### 5.1. Monotherapy with AR Agonists

The systemic treatment of advanced metastatic breast cancer with agonists of the AR initially used steroidal androgens starting in the 1940s, with as many as 15–30% of patients achieving some tumor regression in spite of their unknown hormone receptor status. Androgenic drugs included medroxyprogesterone, testosterone, and fluoxymesterone [87]. Unfortunately, there are undesirable side effects that come with systemic androgen treatment in women that affect other tissues; these include virilization, masculinizing effects, and aggressive behaviors, which in the 1970s, led to steroidal androgen treatments being supplanted with antiestrogen therapeutics such as tamoxifen (reviews in [10,88,89]).

Tamoxifen is a SERM, which means it has differing effects on ER transcriptional activity in different tissues. In breast tissue, tamoxifen is an ER antagonist, but unfortunately, it is an ER agonist in uterine tissue, thereby promoting the proliferation of the endometrium and elevating the risk of uterine cancer. The SERM raloxifen is an antagonist for the ER in breast tissue with negligible effects on uterine tissue and therefore has a lowered risk of endometrial cancer compared to tamoxifen [90]. The success of finding tissue-specific effects with SERMs has led to the development of selective receptor modulators for other steroid hormone receptors, including AR (see Narayanan et al. [91]). The development of SARMs was undertaken in the 1990s with the hopes of maintaining a therapeutic impact similar to testosterone for some types of breast cancer without the unwanted side effects in nontargeted tissues.

The SARM enobosarm (Ostarine, GTx-024, MK-2866) activates AR transcriptional activity and has antitumor activity in the patient-derived xenograft models of AR-positive/ER-positive breast cancer in the presence of estrogen [72]. This is consistent with the SARM RAD140 (EP0062/testolone/vosilasarm), which also inhibits the growth of patient-derived AR-positive/ER-positive xenograft tumors [92]. Importantly, in an AR-overexpressing TNBC cell line (MDA-MB-231-AR), SARMs (enobosarm and GTx-027), but not the AR antagonist bicalutamide, reduced cell proliferation and inhibited the in vivo tumor growth of MDA-MB-231-AR xenografts [93]. There is a possibility that the forced overexpression of AR in TNBC may result in AR acting like a tumor suppressor rather than behaving as an oncogenic driver, which occurs when it is endogenously expressed. The status/expression levels of AR, together with other hormone receptors, are required to determine optimal treatment strategies. Currently, enobosarm is being evaluated as a monotherapy in clinical trials for multiple pathologies, including ER-positive/AR-positive breast cancer as well as AR-positive TNBC breast cancer (NCT01616758, NCT02463032, and NCT04869943). The NCT02368691 trial with endosarm in AR-positive TNBC was terminated due to its lack of efficacy. RAD140 is also being developed for ER-positive/AR-positive/HER2-negative breast cancer and has recently been tested in a phase I/II study (NCT05573126 [94]). Clinical trials targeting AR for breast cancer patients are listed in Table 2.

### 5.2. Monotherapy with AR Antagonists

Therapeutics that inhibit AR transcriptional activity include the use of antiandrogens that bind to the AR-LBD and the ablation of the synthesis of androgens. Preclinical studies have demonstrated that AR inhibitors reduce the tumor growth of some AR-positive breast cancers. The first-generation antiandrogen bicalutamide inhibited the DHT-induced tumor growth of AR-positive TNBC human MDA-MB-453 xenografts in mice [75]. Enzalutamide is a second-generation antiandrogen with improved affinity for AR and increases the survival of prostate cancer patients compared to bicalutamide [95,96,97]. Consistent with bicalutamide, enzalutamide also decreases cell proliferation, increases apoptosis, and reduces the tumor growth of AR-positive TNBC cell lines and xenografts [98].

The clinical testing of enzalutamide in a phase II trial provided evidence that, at 160 mg daily, it was well tolerated in patients with AR-positive TNBC (NCT01889238) [46]. The treatment resulted in an improved clinical benefit rate (CBR) at 16 weeks (25% in all the enrolled patients compared to 33% in the evaluable patients), and a median OS of 17.6 months in the evaluable subgroup compared to a median OS of 12.7 months in all the enrolled patients. Adjuvant treatment with enzalutamide for one year was well tolerated in early-stage AR-positive TNBC, and the one-year DFS was 94% (NCT02750358) [99].

The ablation of androgen biosynthesis by cytochrome P450 17A1 (CYP17A1)-lyase inhibitors such as abiraterone, seviteronel, and orteronel is another approach to reduce AR activity. Abiraterone is administered with prednisone to manage adverse side effects. The 6-month CBR was 20% for AR-positive TNBC patients treated with abiraterone acetate/prednisone (NCT01842321) [100]. This was contrary to its effect on ER-positive breast cancer, where abiraterone induced cell proliferation by activating the ER [101,102]. Seviteronel has dual roles in affecting the AR signaling axis by reducing both androgen biosynthesis and inhibiting AR activity. A phase I study (NCT02580448) showed that seviteronel was well tolerated at 450 mg daily in ER-positive breast cancer and TNBC, and the CBR at 16 weeks was 57% (four out of seven patients) [103]. In contrast, orteronel was tested in a clinical trial for AR-positive metastatic breast cancer where it showed limited clinical activity (NCT01990209) [104].

## 6. AR as a Target in Combinations

### 6.1. Combination Treatments with AR Agonists and Antagonists

A combination of two or more treatments may improve treatment responses for cancer patients by targeting multiple mechanisms of action, thereby reducing the survival and outgrowth of resistant clones. Modulating AR activity either with an AR agonist or AR antagonist with other treatments has been tested in numerous clinical trials for breast cancer patients (listed in Table 3). The majority of combination studies with an AR antagonist tested in clinical trials for breast cancer have been with the antiandrogen enzalutamide. These include the combination of enzalutamide with endocrine therapy, the inhibitors of PI3K, anti-HER2 antibody, and CDK4/6 (cyclin-dependent kinase 4/6) inhibitors. Combinations are being tested in both ER-positive or HER2-amplified breast cancers and TNBC.

### 6.2. Combination Treatments with an AR Agonist

An AR agonist has been tested in clinical trials with pembrolizumab (immunotherapy) as well as abemaciclib (CDK4/6 inhibitor) in TNBC and ER-positive breast cancer, respectively. A phase II clinical trial combining enobosarm and pembrolizumab, an anti-PD-1 (programmed cell death protein 1) antibody, was carried out in AR-positive metastatic TNBC with well-defined levels of hormone receptors: tumors with ER-negative (nuclei staining ≤10%), PR-negative (nuclei staining ≤10%), and AR-positive (nuclei staining ≥10%). This combination was well tolerated, with a CBR of 25% at 16 weeks, but unfortunately, due to the withdrawal of the enobosarm supply, data for analysis were incomplete (NCT02971761) [106]. Enobosarm has also been combined with the CDK4/6 inhibitor abemaciclib, and this is currently in clinical trials for metastatic breast cancer that is ER-positive/HER2-negative/AR-positive, with nuclei staining ≥40% (NCT05065411).

### 6.3. Combination Treatments with an AR Antagonist in ERα-Positive or HER2-Positive Breast Cancer

In ERα-positive or HER2-positive breast cancer, there have been a multitude of clinical trials that have combined an antiandrogen or an inhibitor of androgen synthesis with an ERα-degrader, aromatase inhibitor, or an anti-HER2-antibody. Several clinical trials have tested enzalutamide in combination with endocrine therapy in ERα-positive and/or PR-positive breast cancer and have determined that these are well tolerated (NCT01597193; NCT02953860) [107,109]. Stage 2 of the NCT01597193 trial showed that the CBRs at 16 weeks were 14% and 20% for enzalutamide monotherapy versus the combination of enzalutamide with fulvestrant, respectively, and the CBRs at 24 weeks were 7% and 9%. By contrast, the NCT02953860 study reported that the CBR at 24 weeks was 25.0%, and the median progression-free survival (PFS) was 8 weeks when patients received the combination treatment. Notably, the patients enrolled in the NCT02953860 study had other treatments, including chemotherapy and endocrine therapies, before receiving the combination treatment, and 12 out of 32 patients had prior treatment with fulvestrant. Interestingly, the patients with PFS shorter than 60 days had activated mTOR signaling in their pretreatment biopsies, but patients with PFS longer than 24 weeks had less activated mTOR signaling. Therefore, the addition of an inhibitor targeting PI3K/Akt/mTOR to the combination including AR inhibitors may benefit some patients with metastatic ERα-positive/HER2-negative/AR-positive breast cancer.

A phase II trial evaluating enzalutamide plus exemestane versus exemestane monotherapy (NCT02007512) [108] showed that this combination was well tolerated in patients with ERα-positive and/or PR-positive breast cancer. Although the CBR at 24 weeks was not significantly different between the combination group and the exemestane-alone group, the patients with high levels of AR mRNA and low levels of ESR1 mRNA (ERα) had longer PFS with the combination treatment than with exemestane monotherapy (14.0 months versus 3.8 months). Interestingly, an improved PFS was observed only in patients without prior endocrine therapy, thereby leading to the speculation that patients with prior endocrine therapy may become less dependent on AR or ER activity.

The application of abiraterone acetate/prednisone combined with exemestane in ERα-positive breast cancer patients (NCT01381874) previously treated with nonsteroidal aromatase inhibitors resulted in no improvement in PFS compared to exemestane, which led to the speculation that the lack of clinical benefit was due to abiraterone acetate increasing levels of progesterone [105]. Together, these studies emphasize the potential impact of previous treatments on clinical responses.

A phase II study (NCT02091960) that evaluated the combination of enzalutamide with trastuzumab, an anti-HER2 antibody, in patients with AR-positive/HER2-positive breast cancer showed that this combination was well tolerated, but that there was no significant benefit from the combination [110].

### 6.4. Combinations in TNBC

The fairly recent acknowledgment of the potential of the AR as a driver of AR-positive TNBC has led to the emergence of multiple clinical trials that have tested combination treatments with AR antagonists and inhibitors of cyclin-dependent kinase (CDK) 4/6 and PI3K.

The inhibition of CDK 4/6 by abemaciclib, palbociclib, and ribociclib has been approved by the FDA for the treatment of some breast cancers. AR-positive TNBC or the LAR subgroup is particularly sensitive to CDK4/6 inhibition, and this sensitivity is related to the expression of AR and low levels of cyclin E1 expression [111]. A combination of enzalutamide and palbociclib inhibited the cell viability of AR-positive TNBC, but this inhibition was only observed in RB-proficient cells [112]. The clinical testing of CDK4/6 inhibitors in combination with AR antagonists is ongoing for AR-positive TNBC patients with the combination of ribociclib and bicalutamide (NCT03090165) [113] and palbociclib and bicalutamide (NCT02605486) [114]. Both trials have expected completion dates in late 2024.

Also in TNBC, a combination of AR and PI3K inhibitors was tested in a phase Ib/II trial (NCT02457910) [83]. The evaluable patients receiving enzalutamide plus taselisib reached 35.7% CBR at 16 weeks, although the study was terminated prior to completion due to the limited efficacy of taselisib in metastatic breast cancer in a different trial study that was testing taselisib plus fulvestrant (NCT02340221) [86,115].

Currently, there is an ongoing trial to test enzalutamide plus alpelisib (an α-specific PI3K inhibitor) in AR-positive/PTEN-positive patients with metastatic TNBC or ERα-positive and/or PR-positive/HER2-negative breast cancer (NCT03207529).

### 6.5. Treatments Targeting AR Variants

A major mechanism of resistance that develops in response to AR inhibitors by antiandrogens and inhibitors of androgen synthesis is the expression of constitutively active AR-Vs such as AR-V7 [37,55,56,57,116]. While the majority of the reports are from prostate cancer cells and tissues and CTCs from prostate cancer patients, the expression of AR-V7 is also detected in breast cancer cell lines in response to enzalutamide [39]. These observations suggest that the expression of AR-V7 may be a potential mechanism of acquired resistance to AR inhibitors for breast cancer patients. Clinically, AR-V7 was detected in a TNBC metastatic bone lesion from a patient with recurrent disease after 1 year of adjuvant enzalutamide treatment in spite of AR-V7 not being expressed in the primary tumor in the same patient before enzalutamide treatment [99]. To date, there have been no clinically approved therapeutics that target AR-V7.

Strategies for targeting the AR splice variants include inhibiting AR-NTD and AR-DBD, AR degradation, and AR translation. Examples are listed in Table 4 (as well as illustrated and listed in Figure 1). Most of these inhibitors or modulators have been tested in prostate cancer cell lines, xenograft models, or clinical trials for prostate cancer and not breast cancer. The few exceptions include antisense to AR (AZD5312 or IONIS-AR-2.5Rx), which was reported to have the potential to reduce all forms of AR, including AR-Vs, but all clinical development was halted (clinical trial NCT02144051 completed in 2016). Our lab has developed the first inhibitors of AR-NTD, and these include EPI analogs (anitens), sintokamides, and niphatenones [117,118,119,120,121,122,123]. The EPI and sintokamide compounds specifically inhibited AR-V7 and full-length AR transcriptional activities both in vitro and in vivo [117,118,119,120,121,122]. The EPI compounds have antitumor activity against human xenografts that express the full-length AR and AR-V7. In 2015, our first compound, EPI-506 (ralaniten acetate), entered clinical trials in heavily pretreated prostate cancer patients in whom treatment with enzalutamide, abiraterone, or both had failed (NCT02606123). This marked the first inhibitor of any intrinsically disordered protein to enter clinical trials. While EPI-506 was well tolerated and showed signs of efficacy, its clinical development was halted due to poor pharmacokinetics, which led to an excessive pill burden [124]. A second-generation compound called masofaniten (EPI-7386), with improved pharmacokinetics, entered clinical trials in 2020 for advanced prostate cancer patients (NCT04421222). Masofaniten is also currently being tested in combination with antiandrogens in clinical trials (NCT05075577) based on preclinical reports pointing to the improved efficacy of a combination of EPI-7170 with enzalutamide in castration-resistant preclinical models of prostate cancer [125].

EPI compounds have also shown improved efficacy in combination with taxanes [151]; peptidyl–prolyl isomerase inhibitors [138]; radiation [120]; mTOR inhibitors [152]; and CDK4/6 inhibitors [139]. These studies have typically employed models of prostate cancer; however, a CDK4/6 inhibitor combined with an EPI-7170 was tested in models of human breast cancer. These studies first provided evidence suggesting that EPI-7170 inhibited the growth of AR-expressing SUM159PT human TNBC cells in a dose-dependent manner [139]. EPI-7170 was superior to enzalutamide in disrupting the cell cycle and causing cell cycle arrest in breast cancer cell lines that express AR-V7. The combination of EPI-7170 with palbociclib was superior to monotherapy in altering the cell cycle in AR-positive breast cancer cell lines, reducing the percentage of cells in the S phase, and increasing cell cycle arrest in the G1 phase [139].

## 7. Conclusions and Future Direction

With more and more research on AR signaling in breast cancer, support is accumulating for the idea of AR playing an essential role in some breast cancers. AR has distinct roles in different subtypes of breast cancers. In ER-positive breast cancer, AR behaves as a tumor suppressor, with its function being opposite to ER. In HER2-amplified breast cancer, AR behaves as the oncogenic driver instead of ER. AR also acts as an oncogenic driver in AR-positive TNBC. It has prognostic significance and is a promising therapeutic target to treat some breast cancer patients. Depending on the specific subtype of breast cancer, the modulation of AR expression or its transcriptional activity by either stimulation or inhibition may be an effective therapeutic strategy. Therefore, the expression status of AR and its variants such as AR-V7 should be determined as a standard screening. The determination of hormone levels in addition to the expression status of hormone receptors in breast cancer may contribute to the identification of high-risk subgroups of cancer patients and be useful as a predictor of therapeutic responses. In addition, combination treatment, sequentially or simultaneously, benefits patients by reducing the risk of developing treatment resistance and maximizing the synergistic effects with optimal dosages and potentially reduced side effects. Because of the complexity of breast cancer, triple combination treatments to inhibit or modulate multiple targets may result in greater efficacy compared to the efficacy achieved with just one or two treatments. As the popularity of personalized medicine increases, tools and technologies are needed to efficiently identify or classify specific subtypes of breast cancers with respect to the expression of biomarkers. Artificial intelligence (AI)-based technologies are being developed to support clinicians and scientists in cancer detection, diagnosis, progression, treatment selection, and treatment development. One example in the field of breast cancer is the use of quantitative ultrasound images to create a digital image database for radiomic analysis that may be used for diagnosis or treatment assessment [153]. Another example is the application of network analysis to identify a potential regulatory mechanism and to reprogram breast cancer cells into a specific subtype of breast cancer that is sensitive to targeted therapy [154]. Using AI-based technologies to classify AR-associated breast cancer subtypes based on the AR expression level (or expression pattern) plus the expression of ER, PR, or HER2 will help with the selection of treatments that may effectively benefit cancer patients.

## Figures and Tables

**Figure 1 ijms-25-01817-f001:**
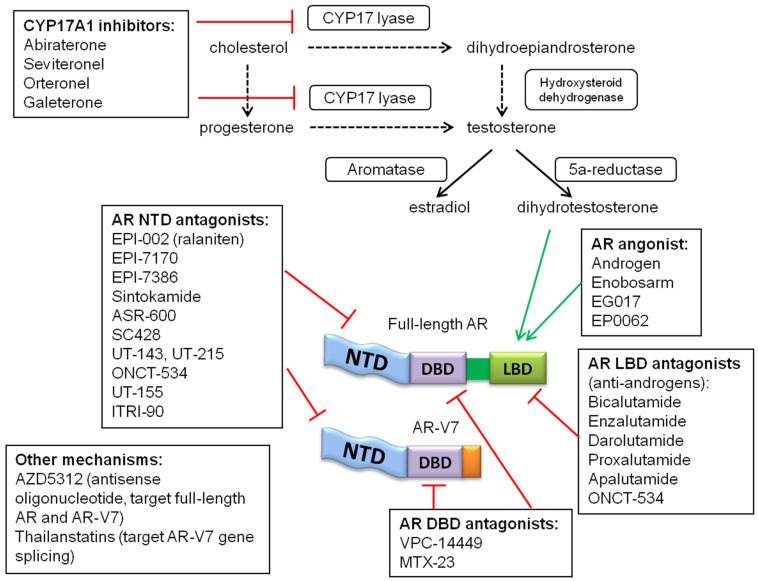
Treatments targeting AR and androgen biosynthesis. Treatments with different mechanisms are shown. NTD, N-terminal domain; DBD, DNA-binding domain; LBD, ligand-binding domain.

**Table 1 ijms-25-01817-t001:** List of antibodies used to detect AR *.

Ab Name or Clone Name	Immunogen	Host	Source
AR27	321 amino acids in the human AR-NTD	mouse mAb	Leica Biosystems (Wetzlar, Germany) (NCL-AR-318)
AR441	AR299-315	mouse mAb	DAKO (Glostrup, Denmark) (M3562), Thermo Scientific (Waltham, MA, USA), Lab Vision (Runcorn, UK), Maixin Biotech (Fuzhou, China)
AR N20	AR1-20	rabbit pAb	Santa Cruz Biotech (Dallas, TX, USA) (sc-816)
AR U407	AR200-220	rabbit pAb	unknown
EP120	unknown	rabbit mAb	ZSGB-BIO (Beijing, China) (ZA-0554)
ER179 (2)	synthetic peptide (unknown location)	rabbit mAb	Abcam (Cambridge, UK) (ab108341)
EPR1535 (2)	synthetic peptide within human AR1-100	rabbit mAb	Abcam (ab133273)
F39.4.1	synthetic peptide of human AR301-320	mouse mAb	BioGenex (Fremont, CA, USA)
SP107	synthetic peptide of human AR300-400	rabbit mAb	Cell Marque (Rocklin, CA, USA), Abcam (ab105225)
AR-V7 **	androgen receptor variant 7	mouse mAb	Precision Antibody (Columbia, MD, USA), (AG-10008)
EPR15656 **	synthetic peptide (human androgen receptor AR-V7-specific peptide)	rabbit mAb	Abcam (ab198394)

* Combined lists from 6 research articles and reviews summarizing AR expression from different studies of breast cancer [5,8,52,58,59,60]. ** Specific antibodies against AR-V7.

**Table 2 ijms-25-01817-t002:** Clinical trials targeting AR or androgen biosynthesis for breast cancer.

Treatment	Direct Target (Mechanism)	Number of Trials as of November 2023 *	Completed **	Ongoing Studies as of November 2023
Abiraterone used with prednisone	CYP17 (a selective and irreversible inhibitor binds to CYP17 to inhibit androgen synthesis)	5	NCT00755885 NCT01381874 NCT01517802 NCT01842321	none
Androgen	AR-LBD (binds to LBD and activates transcriptional activity)	14	NCT00408863 NCT00698035 NCT00725374 NCT01573442 NCT01697345	NCT00080756 (active but not recruiting) NCT05156606 (recruiting)
AZD5312	Antisense oligonucleotide (against AR mRNA for full-length, splice variant, and mutated form of AR)	1	NCT02144051	none
Bicalutamide	AR-LBD (binds to LBD and inhibits transcriptional activity)	11	NCT00468715 NCT02697032	NCT02299999 (active but not recruiting) NCT02605486 (active but not recruiting) NCT03090165 (recruiting) NCT03650894 (active but not recruiting) NCT05095207 (recruiting)
Darolutamide	AR-LBD (binds to LBD and inhibits transcriptional activity)	2	NCT03004534 NCT03383679	none
EG017	AR (SARM)	1	0	NCT05673694 (recruiting)
Enobosarm (GTx-024)	AR (SARM)	8	NCT00467844 NCT01616758 NCT02463032 NCT02746328	NCT02971761 (active but not recruiting) NCT04869943 (active but not recruiting) NCT05065411 (active but not recruiting)
Enzalutamide	AR-LBD (binds to LBD, prevents AR nuclear translocation, and inhibits transcriptional activity)	13	NCT01597193 NCT02953860	NCT01889238 (active but not recruiting) NCT02007512 (active but not recruiting) NCT02091960 (active but not recruiting) NCT02689427 (active but not recruiting) NCT02750358 (active but not recruiting) NCT02955394 (active but not recruiting) NCT03207529 (active but not recruiting)
Orteronel (TAK-700)	CYP17A1 (selective and nonsteroidal inhibitor to CYP17A1)	2	NCT01808040 NCT01990209	none
Proxalutamide	AR-LBD (binds to LBD to inhibit transcriptional activity and downregulates AR expression)	1	NCT04103853	none
RAD140	AR (SARM)	2	NCT03088527	NCT05573126 (recruiting)
Seviteronel (VT-464)	CYP17A1 (selective and nonsteroidal inhibitor to CYP17A1) AR antagonist	3	NCT02130700 NCT02580448	NCT04947189 (recruiting)
SHR3680	AR antagonist	1	0	NCT05928780 (not yet recruiting)

* Any studies related to the treatment and listed on ClinicalTrials.gov (accessed on 24 November 2023) are counted in the number of trials. ** Studies that are not listed or not completed may be terminated, withdrawn, or unknown.

**Table 3 ijms-25-01817-t003:** Clinical trials with combination therapy that include targeting AR.

NCT Number	Treatments	Patients	Results or Status
NCT01381874	Abiraterone acetate/prednisone Abiraterone acetate/prednisone + exemestane (aromatase inhibitor) Exemestane (aromatase inhibitor)	ER^+^ metastatic BC	PFS was not improved (O’Shaughnessy 2016 [105]). (completed August 2018)
NCT02910050	Bicalutamide + aromatase inhibitor (letrozole, anastrozole, or exemestane)	ER^+^AR^+^HER2^−^ metastatic BC	(Unknown status; estimated completion December 2018)
NCT05095207	Bicalutamide + abemaciclib (CDK4/6 inhibitor)	AR^+^HER2^−^ metastatic BC	(Recruiting; estimated completion September 2024)
NCT02605486	Bicalutamide + palbociclib (CDK4/6 inhibitor)	AR^+^ metastatic TNBC	(Active, not recruiting; estimated completion November 2024)
NCT03090165	Bicalutamide + ribociclib (CDK4/6 inhibitor)	AR^+^ TNBC	(Recruiting; estimated completion September 2024)
NCT03650894	Bicalutamide + nivolumab (PD-1 inhibitor) + ipilimumab (CTLA-4 inhibitor)	HER2^−^ BC (including AR^+^ TNBC at screening)	(Active, not recruiting; estimated completion April 2025)
NCT05065411	Enobosarm + abemaciclib (CDK4/6 inhibitor)	ER^+^AR^+^HER2^−^ metastatic BC	(Active, not recruiting; estimated completion January 2024)
NCT02971761	Enobosarm + pembrolizumab (PD-1 inhibitor)	AR^+^ metastatic TNBC	Combination was well tolerated and CBR was 25% at 16 weeks (Yuan 2021 [106]). Active, not recruiting (estimated completion December 2023)
NCT01597193	Enzalutamide Enzalutamide + anastrozole (aromatase inhibitor) Enzalutamide + exemestane (aromatase inhibitor) Enzalutamide + fulvestrant (ER inhibitor)	ER^+^PR^+^ BC	Combination was well tolerated although there were limited efficacy data (Schwartzberg 2017 [107]). (completed January 2018)
NCT02007512	Enzalutamide + exemestane (aromatase inhibitor) Placebo + exemestane (aromatase inhibitor)	ER^+^PR^+^HER2^−^ normal BC	Combination was well tolerated although PFS was not improved (Krop 2020 [108]). (Active, not recruiting; estimated completion December 2023)
NCT02676986	ER^+^ BC cohort: Enzalutamide + exemestane (aromatase inhibitor) Exemestane alone AR^+^ TNBC cohort: Enzalutamide	ER^+^ BC vs. AR^+^ TNBC	(Unknown status; estimated completion March 2020)
NCT02953860	Enzalutamide + fulvestrant (ER inhibitor)	ER^+^HER2^−^ metastatic BC	CBR at 24 weeks was 25% and median PFS was 8 weeks (Elias 2023 [109]). (completed April 2020)
NCT02955394	Enzalutamide + fulvestrant (ER inhibitor) Fulvestrant (ER inhibitor)	locally advanced AR^+^ER^+^Her2^−^ BC	(Active, not recruiting; estimated completion February 2027)
NCT02091960	Enzalutamide + trastuzumab (HER2 inhibitor)	HER2^+^AR^+^ BC	Combination was well tolerated and may offer durable disease control for some HER2^+^AR^+^ patients (Wardley 2021 [110]). (Active, not recruiting; estimated completion December 2023)
NCT02457910	Enzalutamide Enzalutamide + taselisib (PI3K inhibitor)	AR^+^ metastatic TNBC	Combination improved CBR by 35.7% at 16 weeks (Lehmann 2020 [86] published before the study was terminated). (Terminated due to interim analysis showing toxicity; August 2022)
NCT03207529	Enzalutamide + alpelisib (PI3K inhibitor)	AR^+^PTEN^+^ metastatic BC	(Active, not recruiting; estimated completion December 2023)
NCT02689427	Enzalutamide + paclitaxel (microtubule formation stabilizer)	AR^+^ TNBC	(Active, not recruiting; estimated completion December 2025)
NCT02929576	Enzalutamide + paclitaxel (microtubule formation stabilizer) Placebo + paclitaxel (microtubule formation stabilizer)	TNBC	(Withdrawn; estimated completion April 2019)
NCT04947189	Seviteronel-D (seviteronel and dexamethasone) + Docetaxel (microtubule formation stabilizer)	AR^+^ TNBC	(Recruiting; estimated completion December 2024)

BC, breast cancer; ER in this table refers to ERα; PFS, progression-free survival; CBR, clinical benefit rate.

**Table 4 ijms-25-01817-t004:** Clinical trials and preclinical studies of targeting AR variants.

Treatment	Target	Agent Type and Mechanism	Cancer Type (Model or Trial)	Reference
Clinical Trials				
Ralaniten acetate (EPI-506)	AR-NTD	small molecule (AR-NTD inhibitor)	prostate cancer (clinical trial NCT02606123, terminated)	Maurice-Dror 2022 [126]
Masofaniten (EPI-7386)	AR-NTD	small molecule (AR-NTD inhibitor)	prostate cancer (clinical trial NCT04421222, recruiting; clinical trial NCT05075577, recruiting)	Pachynski 2023 [127] Laccetti 2023 [128]
ONCT-534 (GTx-534)	AR-LBD and AR-NTD		prostate cancer (clinical trial NCT05917470, recruiting)	Narayanan 2021 [129]
Niclosamide (PDMX1001)	AR and AR-V7	AR and AR-V7 protein degradation	prostate cancer (cell lines and xenografts) prostate cancer (clinical trial NCT02532114, completed; clinical trial NCT03123978, completed; clinical trial NCT02807805, active, not recruiting)	Liu 2014 [130] Liu 2016 [131] Parikh 2021 [132]
AR-ASO e.g., AZD5312 (IONIS 560131) e.g., ISIS581088	AR mRNA (full-length, splice variant and mutated form)	antisense oligonucleotide ISIS581088 targets intron 1 of mouse AR	AZD5312 in clinical trial (NCT02144051) for solid tumors including breast cancer prostate cancer (cell lines and xenografts) ISIS581088 for prostate cancer mouse model (PTEN KO model)	Yamamoto 2015 [133] De Velasco 2019 [134]
Galeterone (TOK-001)	AR CYP17A1	CYP17A1 inhibitor AR degradation AR inhibition	prostate cancer (cell lines) prostate cancer (clinical trial NCT00959959, completed; clinical trial NCT02438007, terminated; clinical trial NCT01709734, terminated (lack of efficacy))	Yu 2014 [135] Montgomery 2016 [136] McKay 2017 [137]
Pre-Clinical Studies				
EPI-001 Ralaniten (EPI-002)	AR-NTD	small molecule (AR-NTD inhibitor)	prostate cancer (cell lines and xenografts)	Andersen 2010 [117] Myung 2013 [118] Yang 2016 [119]
EPI-7170	AR-NTD	small molecule (AR-NTD inhibitor)	prostate cancer (cell lines and xenografts) breast cancer (cell lines)	Banuelos 2020 [120] Hirayama 2020 [125] Leung 2021 [138] Tien 2022 [139]
Sintokamide (SINT/LPY)	AR-NTD	small molecule (AR-NTD inhibitor)	prostate cancer (cell lines and xenografts)	Sadar 2008 [121] Banuelos 2016 [122]
ASR-600 (analog of Urolithin A)	AR-NTD	small molecule	prostate cancer (cell lines and xenografts)	Chandrasekaran 2023 [140]
SC428	AR-NTD	small molecule	prostate cancer (cell lines and xenografts)	Yi 2023 [141]
ITRI-90	AR-NTD	PROTAC to induce degradation (AR-NTD binding moiety + VHL or CRBN)	prostate cancer (cell lines and xenografts)	Hung 2023 [142]
UT-155	AR AF-1	small molecule (selective AR degrader)	prostate cancer (cell lines and xenografts)	Ponnusamy 2017 [143]
UT-143, UT-215	AR AF-1	small molecule, irreversible covalent binding to Cys	prostate cancer (cell lines and xenografts)	Thiyagarajan 2023 [144]
Thailanstatins	AR-V7	AR-V7 gene splicing	prostate cancer (cell lines and xenografts)	Wang 2017 [145]
CE3-pAM	AR intron 3	morpholino (targets the polyadenylation signal in AR intron 3 (CE3))	prostate cancer (cell lines and xenografts)	Van Etten 2017 [146]
VPC-14449	AR DBD	small molecule	prostate cancer (cell lines and xenografts)	Dalal 2014 [147] Dalal 2017 [148]
MTX-23	AR DBD	PROTAC	prostate cancer (cell lines and xenografts)	Lee 2021 [149]
Dimethylcurcumin (ASC-J9)	AR	AR degradation enhancer	prostate cancer (cell lines and xenografts)	Chou 2019 [150]

## Data Availability

Not applicable.

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
