# Peer review of "Treatments Targeting the Androgen Receptor and Its Splice Variants in Breast Cancer"

_ijms, 2024, doi:10.3390/ijms25031817_

Round 1

Reviewer 1 Report

Comments and Suggestions for Authors

It is well summarized manuscript about targeting AR-V. It is worthy to be published

Comments on the Quality of English Language

Please accept this manuscript

Author Response

Thanks very much to this reviewer. There were no specific concerns raised.

Reviewer 2 Report

Comments and Suggestions for Authors

Dear authors,

I have reviewed the manuscript ijms-2798275. Here are my comments:

Main Question Addressed:

The primary focus of the manuscript appears to be the role of Androgen Receptors (AR) in breast cancer, especially Triple-Negative Breast Cancer (TNBC). It discusses the significance of AR expression, its subgroups, molecular interactions, potential treatment strategies, and clinical trials involving AR-targeted therapies.

Originality and Relevance:

The topic is highly relevant as it addresses TNBC, a subtype known for its aggressiveness and limited targeted treatment options. Exploring AR as a therapeutic target in TNBC, along with its subgroups like the Luminal AR (LAR), is original and valuable. It contributes by highlighting the significance of AR expression, its subtypes, and potential targeted therapies in a subset of breast cancer where treatment options are limited.

Contribution to the Field:

The manuscript provides a comprehensive review of the significance of AR in TNBC, discussing its molecular subgroups, interaction with other signaling pathways (like PI3K/Akt/mTOR), potential therapeutic approaches, and ongoing clinical trials. It adds value by consolidating existing knowledge and recent findings about AR in breast cancer and discussing its implications for treatment.

Methodological Improvements:

While the manuscript is primarily a review and doesn't present original research, it could benefit from deeper insights into the methodological aspects of ongoing clinical trials or experiments discussed. Suggesting potential future directions or experimental designs to further explore AR-targeted therapies in TNBC could enhance its value.

Consistency of Conclusions:

The conclusions drawn seem consistent with the evidence presented in the text. The discussion aligns with the main question of evaluating AR as a potential therapeutic target in TNBC, discussing its different subgroups, molecular interactions, and ongoing clinical trials involving AR-targeted therapies.

Appropriateness of References:

The manuscript seems to include a wide range of references, citing both recent research and foundational studies in the field. 

Author Response

1. Methodological Improvements:

While the manuscript is primarily a review and doesn't present original research, it could benefit from deeper insights into the methodological aspects of ongoing clinical trials or experiments discussed. Suggesting potential future directions or experimental designs to further explore AR-targeted therapies in TNBC could enhance its value.

Response to #1: To address your suggestions under "Methodological Improvements" we have added text describing artificial intelligence-based analysis for AR-related classification and treatments.  These are included in the Conclusions and Future Direction and are provided below.

"Artificial intelligence (AI)-based technologies are being developed to support clinicians and scientists for cancer detection, diagnosis, progression, treatment selection and treatment development. One example in the field of breast cancer is the use of quantitative ultrasound images to create a digital image database for radiomics analysis that may be used for diagnosis or treatment assessment [127]. Another example is the application of network analysis to identify a potential regulatory mechanism and to re-program the breast cancer cells into a specific subtype of breast cancer that is sensitive to targeted therapy [128]. Using AI-based technologies to classify AR-associated breast cancer subtypes based on AR expression level (or expression pattern) plus the expression of ER, PR or HER2 will help with the selection of treatments that may effectively benefit cancer patients."

Reviewer 3 Report

Comments and Suggestions for Authors

Major:

Essential Role in Specific Breast Cancer Types: The primary objective of this review should be to delineate the specific types of breast cancers wherein the androgen receptor (AR) plays a pivotal role. The conclusions briefly touch upon this without any details. Please provide a detailed and specific summary of AR's significance in distinct breast cancer subtypes.

      Utilization of Tools for Identification: The article lacks the use of proper tools for identifying AR-associated treatment for specific types of breast cancer. Knowing the complexity of the subject, incorporating network analysis or artificial intelligence-based software tools could enhance the review's ability to tackle the aim of this review.

·       Figures for Visual Representation: Please consider incorporating figures illustrating receptor structures and variants. Visual aids would significantly enhance the readers' understanding.

·        Balanced Qualitative and Quantitative Descriptions: While qualitative descriptions are predominant, please include quantitative descriptions wherever possible. This balance would augment the review's depth and comprehensiveness.

Minor:

·        Lines 58-61: Please provide appropriate references to support the statements made in lines 58-61 for added credibility.

·        Lines 14-16: Please edit the sentence.

Comments on the Quality of English Language

The quality of English language is fine.

Author Response

Major:

1. Essential Role in Specific Breast Cancer Types: The primary objective of this review should be to delineate the specific types of breast cancers wherein the androgen receptor (AR) plays a pivotal role. The conclusions briefly touch upon this without any details. Please provide a detailed and specific summary of AR's significance in distinct breast cancer subtypes.

Response to #1: Thank you for the comments. A detailed summary of AR roles in different subtypes of breast cancer is included in the section of Conclusions and Future Direction.

2. Utilization of Tools for Identification: The article lacks the use of proper tools for identifying AR-associated treatment for specific types of breast cancer. Knowing the complexity of the subject, incorporating network analysis or artificial intelligence-based software tools could enhance the review's ability to tackle the aim of this review.

Response to #2: Thank you for the comments. A paragraph describing AI-based technology for AR-related classification and treatments for breast cancer is included in the section of Conclusions and Future Direction.

3. Figures for Visual Representation: Please consider incorporating figures illustrating receptor structures and variants. Visual aids would significantly enhance the readers' understanding.

Response to #3: Figure 1 showing AR and AR-V7, and treatments targeting AR was inserted in the manuscript.

4. Balanced Qualitative and Quantitative Descriptions: While qualitative descriptions are predominant, please include quantitative descriptions wherever possible. This balance would augment the review's depth and comprehensiveness.

Response to #4: Thank you for the comments. Quantitative descriptions for some of the published data are now included in the manuscript (total 8 quantitative descriptions are added).

Minor:

5. Lines 58-61: Please provide appropriate references to support the statements made in lines 58-61 for added credibility.

Response to #5: Two references are added.

6. Lines 14-16: Please edit the sentence.

Response to #6: The sentence is edited.

Round 2

Reviewer 3 Report

Comments and Suggestions for Authors

Thank you for adding Figure 1 for an illustrative summary and responding to a few other comments. However, the revision is not appropriate.  The responses to comments #1 and #2, which are two major comments, are incorporated in the section of conclusions and future directions. However, these two major points should be responded to in the main review text. It is inappropriate to see that the authors did not respond to the comments and described the conclusion. They did not include the responses in the review but placed the issue in the future directions.

Comments on the Quality of English Language

FIne.

Author Response

Thank you for reviewing our manuscript. Relevant to our response below to Reviewer 3 are Reviewer 2’s comments, “Main Question Addressed: The primary focus of the manuscript appears to be the role of Androgen Receptors (AR) in breast cancer, especially Triple-Negative Breast Cancer (TNBC). It discusses the significance of AR expression, its subgroups, molecular interactions, potential treatment strategies, and clinical trials involving AR-targeted therapies.” “Consistency of Conclusions: The conclusions drawn seem consistent with the evidence presented in the text. The discussion aligns with the main question of evaluating AR as a potential therapeutic target in TNBC, discussing its different subgroups, molecular interactions, and ongoing clinical trials involving AR-targeted therapies.”

Unfortunately we respectfully feel that Reviewer 3’s comments have already been appropriately addressed in our revision which is consistent with Reviewer's 2 assessments as described in the paragraph above.

Let us further explain. The focus of this review paper, as described in the title (“Treatments targeting androgen receptor and its splice variants in breast cancer”), is about the treatments for breast cancer patients that are related to androgen receptor.

1) In order to provide the rationale for treatments targeting androgen receptor (AR) in breast cancers, we provided section 4 entitled “AR in Breast Cancer” with section 4.1, “Expression of AR” where we describe the expression of AR and its splice variant in breast cancer. There is Section 4.2, “AR roles in ER-positive breast cancer” and section 4.3 entitled, “Elevated levels of androgen in breast cancer”. These sections describe the levels of expression of AR and the levels of its cognate ligand in subtypes of breast cancers. Section 4.4, entitled, “Conflicting consequences of cross-talk between ER and AR” is followed by section 4.5, entitled “AR roles in ER-negative breast cancer”. Our review then built on this foundation to describe the application of agonists and antagonists of AR in the treatment of subtypes of breast cancer as monotherapies or in combinations based upon expression of ER and amplified HER2. Hence in view that we had already provided detailed descriptions of AR’s significance in subtypes of breast cancer, Reviewer 3’s request to “provide a detailed and specific summary of AR’s significance in distinct breast cancer subtypes” was interpreted as a request to add a summary of AR roles in breast cancer in the Conclusions and Future Direction section which we did in the revision. The request to add a detailed and specific summary of AR’s significance in distinct subtypes of breast cancer to the text was already completed, as described above in section 4, and appears to have been overlooked by Reviewer 3.

2) As for Reviewer 3’s comments about network analysis or artificial intelligence-based software tools, we thought it was a nice idea to add in the Future Direction. The placement of this text in Future Directions is fitting because to date there is no approved therapy in breast cancers that targets the AR and because the AR is an emerging prognostic biomarker. It is only proposed that expression of AR be checked as a standard procedure. Artificial intelligence-based tools require machine learning or nowadays deep learning on database. Building a useful database of AR-related treatments will only be possible when the treatments are effective and approved for different types of breast cancers. Hence, Reviewer 3’s statement, “The article lacks the use of proper tools for identifying AR-associated treatment for specific types of breast cancer” seems misplaced for these reasons.

Reviewer 3 first review:

1. The primary objective of this review should be to delineate the specific types of breast cancers wherein the androgen receptor (AR) plays a pivotal role. The conclusions briefly touch upon this without any details. Please provide a detailed and specific summary of AR's significance in distinct breast cancer subtypes.

2. The article lacks the use of proper tools for identifying AR-associated treatment for specific types of breast cancer. Knowing the complexity of the subject, incorporating network analysis or artificial intelligence-based software tools could enhance the review's ability to tackle the aim of this review.

Reviewer 3 second review.

The responses to comments #1 and #2, which are two major comments, are incorporated in the section of conclusions and future directions. However, these two major points should be responded to in the main review text. It is inappropriate to see that the authors did not respond to the comments and described the conclusion. They did not include the responses in the review but placed the issue in the future directions.

Round 3

Reviewer 3 Report

Comments and Suggestions for Authors

The authors did not fully respond to the comments to the original and previously revised versions. 

Comments on the Quality of English Language

Fine.